# A Tale of Two Families: Whole Genome and Segmental Duplications Underlie Glutamine Synthetase and Phosphoenolpyruvate Carboxylase Diversity in Narrow-Leafed Lupin (*Lupinus angustifolius* L.)

**DOI:** 10.3390/ijms21072580

**Published:** 2020-04-08

**Authors:** Katarzyna B. Czyż, Michał Książkiewicz, Grzegorz Koczyk, Anna Szczepaniak, Jan Podkowiński, Barbara Naganowska

**Affiliations:** 1Department of Biometry and Bioinformatics, Institute of Plant Genetics, Polish Academy of Sciences, 60-479 Poznan, Poland; gkoc@igr.poznan.pl; 2Department of Genomics, Institute of Plant Genetics, Polish Academy of Sciences, 60-479 Poznan, Poland; mksi@igr.poznan.pl (M.K.); bnag@igr.poznan.pl (B.N.); 3Department of Genomics, Institute of Bioorganic Chemistry, Polish Academy of Sciences, 61-704 Poznan, Poland

**Keywords:** Fabaceae, *Lupinus*, glutamine synthetase (*GS*), phosphoenolpyruvate carboxylase (*PEPC*), phylogeny, evolution, gene families, duplication/triplication, structural genomics, genome organization, genome evolution

## Abstract

Narrow-leafed lupin (*Lupinus angustifolius* L.) has recently been supplied with advanced genomic resources and, as such, has become a well-known model for molecular evolutionary studies within the legume family—a group of plants able to fix nitrogen from the atmosphere. The phylogenetic position of lupins in Papilionoideae and their evolutionary distance to other higher plants facilitates the use of this model species to improve our knowledge on genes involved in nitrogen assimilation and primary metabolism, providing novel contributions to our understanding of the evolutionary history of legumes. In this study, we present a complex characterization of two narrow-leafed lupin gene families—glutamine synthetase (*GS*) and phosphoenolpyruvate carboxylase (*PEPC*). We combine a comparative analysis of gene structures and a synteny-based approach with phylogenetic reconstruction and reconciliation of the gene family and species history in order to examine events underlying the extant diversity of both families. Employing the available evidence, we show the impact of duplications on the initial complement of the analyzed gene families within the genistoid clade and posit that the function of duplicates has been largely retained. In terms of a broader perspective, our results concerning *GS* and *PEPC* gene families corroborate earlier findings pointing to key whole genome duplication/triplication event(s) affecting the genistoid lineage.

## 1. Introduction

The last decade has seen gradual progress in evolutionary studies on plants, mainly due to simultaneous, rapid advancement in theory, computing, and molecular technology. Legumes, which are the third largest plant family, have attracted the focus of active and collaborative international groups of researchers in the area of systematics and evolution [1,2,3]. Fabaceae, consisting of three major clades—Papilionoideae, Caesalpinioideae, and Mimosoideae—includes important grain, pasture, and agroforestry species that are characterized by an unusual flower structure, podded fruit, and the ability of most species to form nodules with rhizobia [4,5]. Recently, high-quality genome sequences of ten Fabaceae species have been published: *Arachis duranensis*, *Arachis ipaensis* [6], *Cajanus cajan* [7], *Cicer arietinum* [8], *Glycine max* [9], *Lotus japonicus* [10], *Lupinus angustifolius* [11], *Medicago truncatula* [12], *Phaseolus vulgaris* [13], and *Vigna radiata* [14].

Among legume species, due to their outstanding agronomic potential and complex evolutionary history, involving whole-genome duplication [15] and subsequent chromosome rearrangements, *L. angustifolius* has become an object of extensive molecular studies in terms of genomics, proteomics, and metabolomics. Altogether, several thousand molecular markers have been developed, including restriction fragment length polymorphisms (RFLPs), intron targeted amplified polymorphisms (ITAPs), amplified fragment length polymorphisms (AFLPs), molecular fragment length polymorphisms (MFLPs), single sequence repeats (SSRs), expressed sequence tags (ESTs), restriction site associated DNA markers (RADs), and EST-SSRs [16,17,18,19]. Reference genetic linkage maps carrying these markers have been built [17,20,21,22]. As a consequence, sequence-defined markers have been associated with major agronomic traits for this species, including soft seededness, anthracnose and *Phomopsis* stem blight resistance, pod shattering, vernalization requirement, and alkaloid content [16,18,23,24,25,26,27]. Two *L. angustifolius* nuclear genome bacterial artificial chromosome (BAC) libraries have been constructed and almost 15,000 BAC-end sequences have been obtained and annotated [28,29]. Selected BAC clones have been used as anchors for the integration of linkage groups in particular chromosomes by the molecular cytogenetic approach [30,31] and have served as material in evolutionary studies of the *Lupinus* genus [32,33]. Strong microsynteny in gene-rich regions between narrow-leafed lupin and other model legumes has also been observed [17,19,20,34,35]. Moreover, new evidence of widespread triplication within the *L. angustifolius* genome, possibly arising from a polyploidization event, has been found [11]. However, other duplication mechanisms, such as segmental duplications or chromosome additions, from related species cannot be ruled out [36].

Whole genome duplication/triplication and chromosomal rearrangements result in the multiplication of gene content within a particular genome. Gene pairs formed by duplication/triplication usually have a relatively short life span as, due to the relaxed selection constraints, some copies may be lost, others will be pseudogenized, and only a limited number will survive [31,37]. Various factors can alter the size of gene families [38,39,40,41,42,43]. Moreover, the relaxation of selective pressure may have created new developmental opportunities, conferred a selective advantage, and served as an engine for evolutionary changes [44]. Utilizing the explicit reconciliation of gene and species history [45], it is possible to elucidate the optimal sequence of duplication/speciation/loss events under a maximum parsimony framework, as well as derive the topological dating of key events in relation to the reference species tree [46,47]. Taken together, this allows for, as an example, the selection of likely orthologs for investigation as suitable taxonomic markers or for translational studies aimed at understanding neo/subfunctionalization in divergent species.

Taking into consideration the phylogenetic distances and the main characteristics of all legume plants, the most valuable sequences for genetic and evolutionary studies of Fabaceae belong to small gene families which originated early in the tree of life and participate in key enzymatic processes, such as genes encoding glutamine synthetase (GS). *GS* genes are considered to be among the oldest existing and functioning genes in the history of gene evolution [48]. GS is the key enzyme involved in the nitrogen metabolism of higher plants, catalyzing primary ammonium assimilation to form glutamine (GS1—cytosolic GS isoenzyme), as well as the reassimilation of ammonium released by a number of biochemical processes (such as photorespiration, protein catabolism, and deamination of amino acids), and is also related to storage protein accumulation in seeds (GS2—plastid GS isoenzyme) [49]. The central role of GS in nitrogen metabolism in all higher plants is unquestionable. The other gene important in legume evolutionary studies due to its functional correlation with *GS* genes may be phosphoenolpyruvate carboxylase (*PEPC*). PEPC plays a crucial role in the regulation of respiratory carbon flux in vascular plant tissues and green algae that actively assimilate nitrogen. The organic acids supplied by PEPC have several roles within nitrogen metabolism [50]. PEPC proteins are also encoded by a small multigene family with an insufficiently elucidated evolutionary history. However, it is assumed that gene duplication from pre-existing genes, followed by a few amino acid changes and the acquisition of a new gene transcription control, have led to the appearance of new isoforms such as C4 PEPC [51].

Here, we provide characterization of the *L. angustifolius* glutamine synthetase (*GS1* and *GS2*) and phosphoenolpyruvate carboxylase (*PEPC*) gene families, including gene structure determination; genetic localization within narrow-leafed lupin linkage groups (NLLs) and estimations of the *GS1, GS2*, and *PEPC* copy number in the narrow-leafed lupin genome. As sequences of narrow-leafed lupin [11,52] were only available in draft form prior to the start of this study, we decided to combine the screening of the BAC library with available data from genome sequencing. We also address several fundamental questions regarding the evolution of *GS* and *PEPC* gene families in legume plants and 40 other dicots and monocots. We support our evolutionary conclusions with a cross-genera microsynteny analysis of selected genome regions carrying particular *GS* and *PEPC* gene variants in the genomes of narrow-leafed lupin and several legume and non-legume species. Moreover, Fabaceae *GS* and *PEPC* representatives were sampled for selection pressure parameters by both pairwise and branch-site assays.

## 2. Results and Discussion

### 2.1. Narrow-Leafed Lupin GS and PEPC are Encoded by Multigene Families

To tag/select cytosolic *GS* and *PEPC* genes, two sequence-specific probes targeting *GS* and *PEPC* genes, respectively, were amplified and used for narrow-leafed lupin genome BAC library screening. As a result, two BAC clone sub-libraries were created, with BACs representing *L. angustifolius* genome regions carrying *GS* and *PEPC* genes. The presence of analyzed genes within selected BACs was positively verified by PCR amplification and Sanger sequencing with gene-specific primers. The similarity level between particular *GS* and *PEPC* homologs identified in the selected clones was determined. Fragments of analyzed genes (300–400 bp) with a similarity level above 97% were classified as one gene variant and assigned to one contig. Two such contigs and two singletons were constructed for the *GS* sub-library and two contigs with one singleton were constructed for *PEPC*. The composition of the *GS* sub-library is as follows: contig1, clones 015C08 and 087N22; contig2, clones 038E09, 047P22, 088E07, 094A04, and 131H20; and singletons, 036L23 and 059J08. The *PEPC* sub-library contains contig1, clones 067C07 and 083F23; contig2, clones 064J15 and 077K22; and a singleton, 131K15. Taking into consideration these results, the accuracy of BAC library screening with the use of the Southern blot method was calculated to be 50% for both sub-libraries and was considered as being relatively low. It was expected that post-hybridization signals would represent the coverage of the *L. angustifolius* genome in the BAC library [28,53]. The observed phenomenon may reflect the general characteristics of the lupin BAC library and incorporated cloning system used, with the noted instability depending on the carried sequence [28,54,55].

Gene copy number estimation with ddPCR revealed that BAC sub-libraries were lacking some gene duplicates. When the study was initially conceived and the experimental part was conducted, the lupin draft genome had not been officially released. Moreover, the availability of both the scaffold-level [52] genome draft and the latter LupinExpress pseudochromosome-level [11] assemblies has, in some of our other studies, failed to entirely resolve certain areas of the genome, including, for example, the placement of RAP2-7 transcription factor, crucial to alkaloid biosynthesis, reported by Kroc et al. (2019). Therefore, our recent BAC-based study aimed at molecular control of the vernalization response *Ku* locus in the narrow-leafed lupin highlighted a candidate gene (a homolog of FLOWERING LOCUS T) and provided the sequence of the domesticated allele carrying a functional mutation (large indel in the promoter) before the release of the lupin pseudochromosome sequence [25,30]. This finding was later confirmed by genome assembly-based studies. Furthermore, BAC clones may be used as chromosome-specific cytogenetic landmarks for chromosome-scale analysis, as well as for inter-species tracking of conserved chromosome regions and profiling of their structural variation. Both approaches have been used in lupin molecular cytogenetic studies [30,31,32,33]. Indeed, BAC clones from this study (047P22, 036L23, 059J08, 067C07, and 131K15) were recently exploited in parallel research addressing lupin karyotype evolution, providing single-locus anchors for the visualization of chromosomal rearrangements across the panel of ten European and African lupin species. Therefore, even after updating the bioinformatic results to include the newly available genomic data, we decided to retain BAC-derived sequences in the final analysis, both as a record of the train of thought and as valuable supporting evidence directly linking recently developed cytomolecular resources for comparative fluorescent in situ hybridization mapping.

To obtain data on *GS* and *PEPC* genes, sequences of interest were blasted against the narrow-leafed lupin annotated gene set cds v1.0. The search resulted in the identification of nine narrow-leafed lupin *GS* genes in total: seven *GS1* genes (named *GS1a1*, *GS1a2*, *GS1a3*, *GS1b1*, *GS1b2*, *GS1c1*, and *GS1c2*) and two *GS2* genes (named *GS2a1* and *GS2a2*). Nine *PEPC* homologs were identified: *PEPC1a*, *PEPC1b*, *PEPC1c*, *PEPC2a*, *PEPC2b*, *PEPC3a*, *PEPC3b*, *PEPC4*, and *PEPC5* (Table 1). The observed trend in the *L. angustifolius GS* and *PEPC* gene copy number is consistent with the data gathered for other legumes. The *P. vulgaris GS1* gene family contains three active *GS1* genes and one pseudogene [56]. In pea, three active *GS1* genes have been characterized: *GS1*, *GS3A*, and *GS3B* [57]. In *M. truncatula*, two active *GS1* genes (*MtGS1a* and *MtGS1b*), two *GS2* genes *(MtGS2a* and *MtGS2b)*, and one pseudogene (*MtGSc*) were revealed [58]. Two major classes of *GS1* genes have been investigated in *M. sativa* [59]. In the *G. max* genome, there are three *GS1* classes, each represented by at least two functional members [60]. Only one copy of the *GS1* gene was identified in the *A. ipaensi*s and *A. duranensis* species.

According to the proposed evolutionary history of narrow-leafed lupin, it was stated that this species has undergone duplication and/or triplication with several chromosome rearrangements [11,21,36]. Based on a cytogenetic analysis of several species from the *Lupinus* genus, it was also hypothesized that the lupin karyotype has evolved through polyploidy and subsequent aneuploidy [61]. Global analysis of the narrow-leafed lupin transcriptome and legume genome sequence comparative mapping enabled whole genome duplication (WGD) events to be dated. Hane et al. estimated the Papilionoideae radiation at 58 mya with genistoid lineage separation from the other Papilionoideae legumes at 54.6 mya, followed by whole-genome triplication in the genistoid lineage at 24.6 mya [11]. Additionally, the ancient polyploidy event has been confirmed based on an analysis of several genes, such as chalcone isomerases (*CHI*) [62], phosphatidylethanolamine binding proteins (*PEBP*) [30], isoflavone synthetases (*IFS*) [63], and cytosolic and plastid acetyl-coenzyme A carboxylases (*ACCase*) [64]. All listed genes are present in the narrow-leafed lupin genome in multiple variants and evolved by WGDs, evidenced by shared synteny and Bayesian phylogenetic inference. Our results concerning *GS* and *PEPC* gene families support the whole genome duplication/triplication(s) hypothesis.

### 2.2. GS and PEPC Gene Variants are Localized in Distinct Narrow-leafed Lupin Genome Regions

All identified representatives of *GS* and *PEPC* gene families, originating from BACs and in silico genome analyses, were mapped against narrow-leafed lupin genome assembly v1.0, revealing their localization within the analyzed genome. *GS1a1, GS1a2, GS1a3, GS1b1, GS1b2,* and *GS1c2* were assigned to narrow-leafed lupin pseudochromosomes (NLL-04, NLL-16, NLL-14, NLL-11, NLL-09, and NLL-03, respectively), whereas *GS1c1* was assigned to unlinked scaffold11_68. *GS2a1* and *GS2a2* were localized in NLL-19 and NLL-04, respectively. The physical distance between two NLL-04 *GS* genes—*GS1a1* and *GS2a2*—was calculated as approximately 3 Mbp. *PEPC* genes were allocated to nine different NLL pseudochromosomes, as follows: *PEPC1a* to NLL-13, *PEPC1b* to NLL-08, *PEPC1c* to NLL-07, *PEPC2a* to NLL-19, *PEPC2b* to NLL-05, *PEPC3a* to NLL-10, *PEPC3b* to NLL-04, *PEPC4* to NLL-7, and *PEPC5* to NLL-20. Employing the BAC-based results and including those obtained in our previous studies, we provide genomic localization for all identified *GS* and *PEPC* gene variants, as well as the cytogenetic position of four *GS1* and two *PEPC* gene copies in lupin chromosomes. The described gene variants correspond to chromosome-specific cytogenetic markers [31], as follows: *GS1a1*, 047P22_5; *GS1a2*, 087N22_2; *GS1b1*, 036L22_3; *GS1b2*, 059J08_3; *PEPC2a*, 067C07_2; and *PEPC2b*, 131K15_5_3 (Table 1).

In order to resolve the organization of multiple genome regions carrying distinct sequence variants of *GS* and *PEPC*, narrow-leafed lupin genome regions carrying these genes were extracted from the assembly and, together with seven sequenced BAC clone inserts (three with the *PEPC* genes 064J15, 067C07, and 131K15, and four with the *GS* sequences 036L23, 047P22, 059J08, and 087N22), were annotated with putative functions. BAC sequences were mapped onto narrow-leafed lupin scaffolds and selected regions were truncated into a uniform length of 100 Mbp. Four scaffolds remained with the original lengths: scaffold192, 88,054 bp; scaffold11_68, 28,507 bp; scaffold9_1, 31,494 bp; and scaffold59_19, 103,921 bp. Analysis revealed the average GC content of 32.95% and 33.23% for *GS* and *PEPC* regions, respectively. The observed occurrence of repetitive elements in genome fragments flanking *GS* and *PEPC* gene variants varied from 0% (*GS1b1*, scaffold73) to 15.63% (*GS1a2*, scaffold106), and from 1.17% (*PEPC1c*, scaffold274) to 15.64% (*PEPC3b*, scaffold296), with retrotransposons (Ty1/Copia and Gypsy/DIRS1) and DNA transposons (DNA/Mule-MuDR type) being the most abundant.

It has been confirmed that the narrow-leafed lupin genome is highly repetitive (57%) [11], with well-organized gene-rich regions. In addition to satellites sensu lato, long terminal repeat (LTR) retrotransposons and DNA transposons were revealed as the most common, with only a small proportion of non-coding RNA [11,19,31,65]. Due to the “copy and paste” mechanism underlying the amplification of LTR retrotransposons, they have been shown to make up the largest classes of transposable element (TE) content in the genomes of most flowering plants, greatly contributing to increases in size of their host genome [66]. As reported in studies concerning *Arabidopsis*, soybean, and flax genomes, *Copia* elements are largely located within and/or close to gene-coding regions, which suggests that these elements may have the dominant influence on the evolution of some gene families [67,68,69]. Gene prediction revealed features characteristic of gene-rich regions, with an average of 13 coding sequences per 100 Mbp for both *GS* and *PEPC* gene regions (Table 1, Appendix A). The obtained data for the frequency of coding sequences within analyzed regions of the narrow-leafed lupin genome showed a lower coding sequence (CDS) abundance than in our previous studies [19,31]. This low number of genes in *GS1a2*, *GS2a1*, and *PEPC3b* neighborhoods is primarily due to the high content of repetitive elements in the surrounding regions.

### 2.3. GS and PEPC Gene Variants Present a Conserved Sequence Structure among All L. angustifolius Homologs and Other Legume Species

To investigate the structural changes of the *GS* and *PEPC* genes, sequence data from 46 species originating from 26 plant families were gathered (Appendix A). In total, 244 sequences of *GS* homologs were subjected to exon/intron determination. The average CDS length for *GS1* (178 sequences analyzed) was established as 3259 bp, with 12 exons as the dominant structure, whereas for *GS2* (46 sequences analyzed), the value was 3866 bp, with 13 exons. Legume GS homologs (36 sequences of *GS1* and *GS2*) presented a conserved gene structure consistent with the pattern described above. Indeed, only the structures of four *GS1* genes were different: Lj0g3v0335159 from *L. japonicus—*nine exons; TR_5g077950 from *M. truncatula—*nine exons; gene13764 (LOC107631250) from *A. ipaensis—*10 exons; and GLYMA02G41106 from *G. max—*10 exons. In the case of *GS2* homologs, only gene3699 (LOC107637831) from *A. ipaensis* with 14 exons and Tp57577_TGAC_v2_gene28916 from *Trifolium pratense* with 20 exons showed an atypical gene structure (Appendix A).

To establish the structure of *PEPC* gene family representatives among higher plants, 223 sequences were analyzed. Based on the exon/intron organization, two groups were formed. The first group contained 167 sequences with an average length of 5645 bp (min. 3102 bp, max. 17,375 bp) structured into 10 exons. Nevertheless, some variation in exon composition was found, particularly in the sequences GSMUA_Achr9G06420_001 from *Musa acuminata* and MDP0000258440 from *Malus domestica*, consisting of 17 and 19 exons, respectively. The second group carried 57 sequences with an average length of 9268 bp (min. 4144 bp, max. 26,587 bp), mainly organized into 18–24 exons (mode value 20). Sixty-four sequences originating from the Fabaceae family presented very low variation in sequence organization. Only MTR_8g463920 and MTR_0002s0890 from *M. truncatula*, gene 1498 (LOC101500264) and gene 3089 (LOC101497901) from *C. arietinum*, and Tp57577_TGAC_v2_gene11496 from *T. pratense* showed differences in the gene structures (Appendix A).

The structures of all identified *L. angustifolius GS* and *PEPC* genes were established. The *GS* sequence lengths varied from 3550 to 8730 bp for *GS1* homologs and from 4002 to 4890 bp for *GS2*. Coding sequence organization was highly conserved within *GS1* (12 exons) and *GS2* (13 exons) groups, despite the observed dissimilarities in lengths. CDS lengths were as follows: *GS1a*, 1071 bp (356 aa); *GS1b1*, 1071 bp (356 aa); *GS1b2*, 1062 bp (353 aa); *GS1c*, 1074 bp (357 aa); and *GS2a1* and *GS2a2*, 1299 bp (432 aa). A major structural difference in *GS* genes was observed for *GS1b2*, where exon number 12 was significantly shorter than in other homologs (144 vs. 153 bp, respectively). Moreover, 5′ and 3′ *GS* regulatory regions revealed high variation between all analyzed sequences, both in length and composition. *PEPC* genes were organized into 10 exons, and the coding sequence length varied from 2901 to 2907 bp (from 966 to 968 aa), excluding *PEPC5*, which had a 3135 bp (1044 aa) length structured into 20 exons and thus significantly deviated from the other *PEPC* sequence variants. The observed level of sequence similarities within the *PEPC* clade is considered as being high. However, major differences in the length and composition of 5′ and 3′UTR regions were noted (Appendix A).

### 2.4. The Initial GS and PEPC Complement was Subsequently Duplicated in a Lineage-Specific Manner and Can be Traced to the Common Ancestor of Legumes

The reconstructed phylogeny of plant *GS* genes yielded several insights with regards to legume enzymes. Firstly, the initial representation of this family in Fabaceae is shown to have consisted of three ancestral clades (Figure 1, Figure 2, and Appendix A) for a simplified phylogenetic tree of relationships. The first monophyletic clade (denoted as *GS2—*Table 2) encompasses the known types of *GS2* loci, which are annotated as chloroplastic proteins encoded in the nuclear genome. Duplicates are present in multiple, rather than singular, cases of divergent legumes and were previously found to be expressed in seeds, at least in the case of *M. truncatula* [70]. The other two clades (*GS1cs1* and *GS1cs2*) carry genes encoding cytosolic proteins corresponding to cytosolic isoforms preferentially expressed in different organs/at different developmental stages (i.e., *GS1cs2—α*, and *GS1cs1—β* and *γ* subunits described in early comparative analyses [71]). The placement of *Vitis vinifera* and multiple malvid sequences between the two clades points to the *GS1cs1*/*GS1cs2* ancestral split either coinciding or shortly following the γ triplication common to both rosids and asterids [72]. Additionally, the *GS1cs1* ancestral split, which resulted in the separation of β and γ subclades (constitutively expressed vs. nodule enhanced, respectively), is shown to have occurred early in the evolution of legumes (possibly prior to the separation of genistoid lineage, with *GS1cs1-β* encoding loci seemingly not having been retained in the NLL reference genome).

Both one *PEPC2* (PTPC, plant-type PEPC [50]) clade and two *PEPC1* (BTPC, bacterial-type PEPC) clades can be clearly characterized as monophyletic in legumes. Therefore, three ancestral genes inherited from an early rosid are indicated, each of which was duplicated prior to the divergence of genistoid/dalbergioid lines and can be traced to the common ancestor of legumes (*PEPC1a, PEPC1b, PEPC2—*see Table 3 for a full summary and Figure 3, Figure 4, and Appendix A for relevant fragments of phylogenetic reconstruction). The ancestral duplication giving rise to *PEPC1a* and *PEPC1b* legume plant-type *PEPC* subgroups likely dates back to core eudicots (coincident with γ triplication or closely following the event). An additional legume-specific duplication event is implied in *PEPC1b*, although incomplete lineage sorting artefacts cannot be ruled out. Indeed, as with available reconstructions of legume phylogeny based on housekeeping genes, the ordering of early diverging dalbergioid and genistoid lineages is seen to alternate between two possibilities.

The initial *GS1* complement was subsequently duplicated in a lineage-specific manner and available evidence (including intact intron-exon structure, which is prior published evidence in the case of alfalfa and common bean) indicates that the functionality of these duplicates has been largely retained in extant crop species. In regard to lupin, the narrow-leafed lupin enzymes are shown to be the result of such duplications and are thus paralogous to the closest counterparts from non-genistoid groups. As a closing side note, the overall resolution of events on the basis of the phylogeny (evolution of cytosolic GSI-encoding genes) suggests that monocot family members might be more ancient than dicot ones, stemming from the selective culling of duplicates predating the separation of both lineages (in line with the split between cytosolic and plastid eukaryotic GS, likely predating monocot/dicot divergence) [48]. However, it is worth noting that the resolution of these basal events lacks the support necessary to make strong inferences (less than 50% bootstrap probabilities for consensual clades).

Analogous to the *GS* case, most of the retained *PEPC* duplications are late and species-specific (as seen in the soybean, lotus, and lupin genomes). In this case, the reconciled *PEPC* phylogeny supports most lupin gene family members being late paralogs (*PEPC1a.2* and *PEPC1b.1—*single duplication, and *PEPC1b.2—*either two rounds of duplication and loss or triplication in the lineage). The inference of possible subsequent duplications/triplication (both here and in the GS1cs1 γ clade) corroborates the earlier findings, pointing to events affecting the genistoid lineage [36].

### 2.5. Compared to GS Genes, the History of Coding Sequences of PEPC Genes More Closely Recapitulates the History of Species

A maximum likelihood codon-based phylogenetic species tree of 46 reference plant genomes, based on 29 putative single-copy orthologs with the best coverage and uniqueness, was generated in order to track species evolution. The obtained species phylogeny (Figure 5) is highly supported, with only two major differences from the accepted consensus (e.g., The Angiosperm Phylogeny Group 2016). One of these is the alliance of lycopod *Selaginella* and moss *Physcomitrella.* The grouping of these lineages is likely an artefact of rapid diversification in early land plant lineages and could be observed in PEPC/GS phylogenies. Additionally, a significant observed difference is the grouping of *Citrus sinensis* (malvid, order *Sapindales*) with representatives of the rosid order *Malpighiales* (*Ricinus communis*, *Populus trichocarpa*, and *Salix purpurea*). Notably, the phylogeny of the latter order has still not been entirely resolved, with the whole COM (*Celastrales*, *Oxalidales*, and *Malpighiales*) clade placement in rosids being challenged by different datasets [73]. Otherwise, the support for consensus topology is strong and the relationships, in particular the topology of the legume clade, support the earlier consensus [74,75].

Primary metabolism genes were frequently good candidates for molecular taxonomic markers, provided that paralogy was taken into account and suitable low/single copy orthologs were chosen for inference [76]. In this context, the members of *GS* and *PEPC* subfamilies were considered as good candidates in the past. Our results do not fully corroborate these findings.

Contrary to early inquiries [4,77], chloroplastic glutamate synthetases are not particularly good taxonomic markers for legumes. The *GS* phylogeny clearly confirms the existence of multiple, functional copies and the reconstructed ancestry contains both late duplications (*L. angustifolius*, *M. truncatula*, *L. japonicus*, and *G. max*) and traces of earlier events (e.g., positioning *L. angustifolius* sequences, which implies early duplications). From the point of view of future studies, *PEPC* clades provide better candidates for supplementary markers (bacterial-type *PEPC* sequences from clades α and β), as there are less duplications and the phylogenetic signal is strong (as exemplified by the bootstrap support of inner bipartitions). This is supported by past findings demonstrating that WGD may have played a lesser role in the evolution of the *PEPC* family in land plants [78]. However, in all (recent) cases, paralogy should be taken into account (e.g., by targeting UTR regions in order to distinguish paralogs).

More interestingly, the general patterns of lineage-specific duplications suggest that sub-functionalization and/or regulatory rewiring played a large role in shaping the extant carbon and nitrogen primary metabolic pathways in some lineages (*L. angustifolius*, *L. japonicus*, and *G. max*). This is also corroborated by the conserved gene structure and further analyses of selection pressure, which show a lack of changes in core ligand-interacting residues of the encoded proteins. Taken together, the evidence points to regulatory rather than mechanistic changes driving the diversification of both *GS* and *PEPC* family members. Whether this is a result of the differential retention of functional duplicates or different frequency of events, the outcome remains pertinent for future translational/comparative studies of legumes and merits more investigation.

### 2.6. L. angustifolius Genome Regions Carrying GS and PEPC Genes Arise from Duplication/Triplication with Additional Complex Chromosome Rearrangements

*Lupinus angustifolius* genome regions carrying all identified variants of *GS* and *PEPC* genes were subjected to comparative mapping to nine well-defined legume genome assemblies. Several patterns of sequence collinearity in these loci were identified. In particular, a high level of microsynteny was observed for the region carrying *GS1a1* and *A. duranensis* chromosome 3 (122.31 Mbp), *A. ipaensis* chromosome 3 (122.88 Mbp), *C. arietinum* chromosome 6 (0.61 Mbp), *C. cajan* chromosome 1 (4.3 Mbp), *G. max* chromosomes 11 (30.88 Mbp) and 18 (3.47 Mbp), *L. japonicus* chromosome 6 (3.75 Mbp), *M. truncatula* chromosome 3 (2.94 Mbp), *P. vulgaris* chromosome 1 (49.04 Mbp), and *V. radiata* chromosome 3 (9.32 Mbp). All these regions carry (at least) one copy of the *GS1* sequence. The narrow-leafed lupin region containing gene *GS1a2* revealed collinearity links to the same regions as those characterized for *GS1a1*, suggesting the occurrence of lineage-specific duplication. A more complex pattern was observed for *GS1b1* and *GS1b2* regions. Well-preserved sequence collinearities of these regions to loci at *A. duranensis* chromosome 7 (14.10 Mbp), *A. ipaensis* chromosome 7 (15.23 Mbp), and *C. cajan* chromosome 2 (8.45 Mbp), which do not carry any (even considerably truncated) *GS* gene sequences, were observed. This may indicate that some *GS1b* gene copies were eliminated during the evolution of these species. Moreover, two *GS1b* sequence variants matched one region of *V. radiata* chromosome 6 (7.14 Mbp), *P. vulgaris* chromosome 8 (55.14 Mbp), and *G. max* chromosomes 2 (43.20 Mbp) and 14 (47.82 Mbp) with a high level of sequence similarity. These regions encode *GS* sequences. *GS1c1* regions did not reveal conserved synteny among any of the species analyzed, only showing alignments between *GS* gene sequences. *GS1c2* regions yielded high collinearity alignments to loci carrying corresponding *GS* sequences at *A. duranensis* chromosome 5 (96.66 Mbp), *A. ipaensis* chromosome 5 (129.41 Mbp), *C. arietinum* chromosome 8 (11.79 Mbp), *C. cajan* scaffold 132405, *G. max* chromosomes 7 (10.08 Mbp) and 9 (39.77 Mbp), *L. japonicus* chromosome 2 (10.53 Mbp), *M. truncatula* chromosome 6 (26.24 Mbp), *P. vulgaris* chromosome 4 (42.89 Mbp), and *V. radiata* chromosome 1 (8.22 Mbp).

In the case of *GS2* regions, clear evidence of sequence collinearity was observed in all analyzed legumes: *A. duranensis* chromosomes 1 (97.70 Mbp) and 4 (3.66 Mbp), *A. ipaensis* chromosomes 1 (128.24 Mbp) and 4 (4.93 Mbp), *C. arietinum* chromosome 1 (4.92 Mbp), *C. cajan* chromosome 2 (8.45 Mbp), *G. max* chromosomes 13 (32.46 Mbp) and 15 (7.96 Mbp), *L. japonicus* chromosome 6 (20.97 Mbp), *M. truncatula* chromosome 2 (7.20 Mbp), *P. vulgaris* chromosome 6 (26.87 Mbp), and *V. radiata* chromosome 10 (16.06 Mbp).

To summarize, all legume regions carrying at least one copy of the *GS* gene revealed shared synteny (Figure 6) to at least one narrow-leafed lupin region carrying a corresponding homologous copy. Some of them matched duplicated regions in the narrow-leafed lupin genome located on different chromosomes and carrying different homologous gene copies, providing clear evidence of ancient duplications of chromosome segments that did not result in the further elimination of additional gene copies.

The set of legume regions carrying *PEPC* genes had more complex patterns of collinearity links. Two types of syntenic relationship were observed, related to regions carrying a *PEPC* gene and to regions lacking such a gene. Moreover, numerous local duplications in the analyzed data set were revealed. Highly conserved microsynteny, expressed by high values of the total score of sequence alignments, was observed for *PEPC1a*, *PEPC1b*, *PEPC1c*, and *A. duranensis* chromosomes 3 (26.99 Mbp) and 7 (72.62 Mbp); *A. ipaensis* chromosomes 3 (29.54 Mbp) and 8 (27.74 Mbp); *C. arietinum* chromosome 1 (47.88 Mbp) and scaffold 1545; *C. cajan* chromosome 10 (12.46 Mbp) and scaffold 380; *G. max* chromosomes 6 (35.35 Mbp), 12 (29.90 and 38.94 Mbp), and 13 (37.24 Mbp); *L. japonicus* chromosome 3 (3.90 and 14.19 Mbp); *M. truncatula* chromosomes 2 (32.09 Mb) and 8 (22.56 Mbp); *P. vulgaris* chromosomes 5 (10.24 and 19.00 Mbp) and 11 (42.06 Mbp); and *V. radiata* chromosomes 2 (16.95 and 21.00 Mbp) and 5 (35.59 Mbp). All these regions carry *PEPC* gene sequences. *PEPC2a* and *PEPC2b* revealed high collinearity links to the same legume genome regions as *PEPC1a*, *PEPC1b*, and *PEPC1c*. *PEPC3a*, *PEPC3b*, and *PEPC4* genes showed conserved synteny to regions carrying *PEPC* homologs located at *A. duranensis* chromosomes 3 (21.75 Mbp) and 8 (33.03 Mbp); *A. ipaensis* chromosomes 2 (7.18 Mbp) and 3 (24.05 Mbp); *C. arietinum* chromosomes 1 (12.63 Mbp) and 6 (21.32 Mbp); *C. cajan* scaffolds 293 and 330; *G. max* chromosomes 6 (46.94 Mbp), 12 (36.95 Mbp), and 13 (39.10 Mbp); *M. truncatula* chromosome 4 (30.90 Mbp); *P. vulgaris* chromosomes 5 (28.48 Mbp) and 11 (29.34 Mbp); and *V. radiata* scaffold 23. The *PEPC4* region also had highly conserved synteny to some regions lacking *PEPC* sequences, namely *A. ipaensis* chromosome 8 (12.40 Mbp), *C. cajan* chromosome 4 (9.64 Mbp), *G. max* chromosome 12 (13.31 Mbp), *L. japonicus* chromosome 3 (35.10 Mbp), and *V. radiata* scaffold 149. This may suggest that *PEPC4* gene copies were removed from these regions during evolution. In the *PEPC5* region, no microsynteny was found between lupin and other legumes. Nevertheless, several orthologs of *PEPC5* were described. In general, *PEPC* genes revealed complex patterns of microsynteny, indicating both lineage-specific and ancestral duplications, as well as possible deletions of excessive gene copies (Figure 7, Appendix A). The distribution of collinearity links provided a clear line of evidence that both *GS* and *PEPC* gene families have expanded in legumes through segmental duplications, which may be considered as landmarks of two ancient WGD events.

### 2.7. The Major Events Promoting the Evolution of GS and PEPC Genes in Legumes were Whole-Genome Duplications

It is a well-accepted hypothesis that the evolution of legumes has been driven by an ancient WGD event which putatively occurred in the progenitor line of Papilionoideae about 50–65 mya, providing the tetraploid ancestor and launching the divergence of ancient lineages of Papilionoideae [3,8,75,79,80]. Traces of that event have been identified in numerous clades spanning the legume tree of life, from Xanthocercis and Cladrastis through dalbergioids (*Arachis* spp.) and genistoids (e.g., *L. angustifolius*), to more recent lineages of millettioids (*P. vulgaris*, *G. max*, *C. cajan*, and *V. radiata*) and galegoids (*M. truncatula*, *L. japonicus*, and *C. arietinum*) [1,3,74,80,81]. Some species have retained relatively large numbers of ancient tetraploid regions (i.e., 309 regions in *M. truncatula* carrying 4198 genes or 343 regions in *G. max* with 9486 genes). Taking into consideration the topology of the legume *GS1c1* tree, this ancestral duplication might have contributed to the origin of β and γ subclades. A similar explanation might be proposed for the emergence of α and β groups of *PEPC1a*, *PEPC1b*, and *PEPC2*, supported by both phylogenetic inference and the synteny-based approach. However, the lack of genome sequencing data for early diverging legumes hampers such a comprehensive comparative analysis and precludes drawing firm conclusions.

During the early divergence of some downstream lineages, dated to roughly ~30–55 mya, additional independent WGD events probably occurred, affecting Mimosoideae-Cassiinae-Caesalpinieae, Detarieae, Cercideae, and *Lupinus* clades [75]. Large-scale duplication and/or triplication in the *L. angustifolius* genome has been well-evidenced by recent studies involving linkage and comparative mapping [17,36] and microsynteny analysis of selected gene families [30,31,34,62,63]. These WGD events apparently contributed to multiplication of the gene copy number of *L. angustifolius GS* and *PEPC* genes because hypothetical duplicates were found in sister branches of the phylogenetic tree and the genome regions harboring these genes shared common collinearity links. Some lineages experienced WGD events relatively recently, including soybean (~13 mya), carrying numerous genes in the duplicated state [3,9]. All *GS* and *PEPC* subclades, except for *PEPC1a-α*, were shown to carry hypothetical survivors of such an event. Hypothetical legume tandem duplicates were only identified in the *GS* family: in *P. vulgaris*, *V. radiata*, and *G. max* for *GS1cs1* and *L. japonicus* and *M. truncatula* for *GS2*. This is an expected outcome, as tandem duplication has been suggested to be a typical mechanism for the expansion of genes, representing flexible steps in the biochemical pathways or located at the end of pathways, where they do not affect many downstream genes [82]. *GS* and *PEPC* are genes encoding key enzymes involved in crucial metabolic pathways. Therefore, the appearance of additional copies without duplication of the whole pathway might have been selected against by evolutionary processes. On the contrary, the WGD event copies the entire molecular machinery, enabling the further evolution and divergence of redundant networks [83]. Moreover, the type of duplication contributes to the further evolutionary fate, demonstrated by different gene expression patterns and the methylation status of duplicates [84]. A recent expression quantitative trait loci mapping study of an *L. angustifolius* recombinant inbred line population (83A:476 x P27255) provided leaf transcriptomic profiles for 30,595 genes, including all *GS* and *PEPC* homologs present in the genome, except *GS2a2* unannotated hitherto [85]. Gene expression values corresponding to *GS* and *PEPC* homologs were extracted from the Appendix A, Table 6, of Plewiński et al. study [85] and are presented here in Table 4 for direct reference. Indeed, that survey highlighted significant differences in leaf expression levels between particular gene duplicates, namely between *GS1a1* and *GS1a2* or *GS1a3* (43.1 ± 16.4 vs. 13.8 ± 5.2 and 11.5 ± 5.0, respectively); *GS1b1* and *GS1b2* (0.3 ± 0.3 vs. 2.6 ± 1.2, respectively); *GS1c1* and *GS1c2* (0.1 ± 0.2 vs. 187.5 ± 52.4, respectively); *PEPC1a*, *PEPC1b*, and *PEPC1c* (17.0 ± 4.0 vs. 0.5 ± 0.5 vs. 65.0 ± 8.7, respectively); and *PEPC3a* and *PEPC3b* (10.5 ± 2.5 vs. 51.0 ± 11.2, respectively) [85]. The observed differences in the gene expression of *L. angustifolius GS* and *PEPC* paralogs support the previously mentioned hypothesis on the expected sub-functionalization of WGD-derived duplicates.

### 2.8. The Majority of Positively Selected GS and PEPC Genes are Duplicates

According to the topology of the majority of consensus trees, 85 pairs of duplicated legume *GS* and *PEPC* sequences were selected, including those located in sister branches and those originating from different subclades (if applicable). The analysis of the nonsynonymous to synonymous substitution rate (Ka/Ks) ratio revealed that all pairs except for Lj6g3v1887800/Lj6g3v1953860 and Lj6g3v1887790/Lj6g3v1953860 were under strong purifying selection, with Ka/Ks values ranging from 0.00 to 0.32 (Appendix A). The two gene pairs mentioned above had a neutral (Ka/Ks) ratio (0.87). The average Ka/Ks ratio was similar in all species except *L. japonicus*: namely 0.09 in *A. ipaensis* and *V. radiata*; 0.10 in *P. vulgaris*; 0.11 in *C. arietinum* and *G. max*; 0.12 in *T. pratense*; and 0.13 in *C. cajan*, *M. truncatula*, and *L. angustifolius*. The outlier value calculated for *L. japonicus* (0.29) resulted from the two sequence pairs with neutral ratios mentioned above. The average Ka/Ks ratio differed between gene clades, from 0.07 to 0.08 in *PEPC1*a and *PEPC1b*, through 0.12 to 0.15 in *GS1_cs2*, *PEPC2*, and *GS1_cs1*, to 0.32 in *GS2* (0.10 in *GS2* without two *L. japonicus* sequence pairs under neutral selection). To address the selection pressure in a wider phylogenetic context, a branch-site test of episodic positive selection was performed for monophyletic clades, as well as all branches, for particular legume species (Appendix A). Of the 163 combinations studied, statistically significant signals of positive selection were revealed for 16 foreground branches; namely, five for *GS1_cs1*, four for *GS2*, three for *PEPC2*, two for *PEPC1a*, and single branches for *GS1_cs2* and *PEPC1b*. *L. japonicus* and *A. ipaensis* revealed the highest number of branches putatively affected by positive selection: four and three, respectively. *C. arietinum* and *T. pratense* revealed two branches with positive selection markers, whereas *C. cajan*, *G. max*, *L. angustifolius*, *M. truncatula*, and *V. radiata* showed only single branches with such residues. Different amino acid positions were altered and no common pattern for any gene clade was observed.

The majority of positively selected genes were duplicates (13 vs. 3). Duplicates revealed common selection patterns for *A. ipaensis* (*GS2* and *PEPC2*) and partially similar patterns for *L. japonicus GS2*. This may indicate that episodic positive selection occurred in these lineages before duplication events. No correlation between the inferred type of duplication (local vs. WGD) and selection pressure parameters was found; remnants of positive selection were found in both types of duplicates.

Amino acid positions altered by relaxed selection constraints did not include known ligand interacting sites (ATP, glutamate, ammonia, and metal coordination sites were evaluated according to [86]). However, few sequences were considerably truncated and lacked several ligand binding sites, namely: *GS1_cs1*, Lj0g3v0335159 and GLYMA02G41106; *GS1cs2*, Lj2g3v0658180; and *GS2*, Lj6g3v1953860.

Calculated Ka/Ks values highlighted the high selection pressure acting on GS and PEPC paralogs. In general, selection constraints are related to the position of the enzyme in metabolic pathways, as well as the contribution of performed enzymatic activity for basic cell metabolic networks. Usually, genes encoding enzymes located at the top of the metabolic pathway are under stronger purifying selection than downstream ones [87]. An association between the selective pressure acting on a gene and the position of an encoded enzyme in the pathway was revealed in a wide metabolic context [88,89], including *L. angustifolius* genes encoding isoflavone synthase and acetyl-coenzyme A carboxylase [63,64]. A higher selection pressure acts on central and highly connected enzymes, enzymes with high metabolic flux, and enzymes catalyzing reactions that are difficult to bypass through alternative pathways [88]. Moreover, enzymes participating in primary metabolism are usually under a constant strong selective pressure, whereas enzymes performing specified metabolism are under weaker negative selection [89]. One of the postulated explanations for the above pattern is that these specified metabolism genes initially experienced positive selection (higher rate than primary metabolism genes) [90].

## 3. Material and Methods

### 3.1. Research Material

This study was carried out with the use of *L. angustifolius* cv. Sonet germplasm obtained from the Polish Lupin GenBank in the Breeding Station Wiatrowo (Poznań Plant Breeders Ltd., Wiatrowo, Poland) and the narrow-leafed lupin genome BAC library [28].

### 3.2. Identifying GS and PEPC in the L. angustifolius Genome

*GS* and *PEPC* gene models were prepared on the basis of available data on legumes and used as anchors of gene-specific probes. Exon/intron numbers and lengths and elements conserved among several legumes were determined. Accessions AC174349.23 (*M. truncatula*) and L39371.2 (*M. sativa*) served as templates for *GS1* and *PEPC* gene-specific primer design, respectively. The PCR amplification was performed with the use of *L. angustifolius* genomic DNA as a template (25 ng DNA), Taq polymerase (Novazym, Poznan, Poland) supplied with 1× PCR buffer and 2.5 mM Mg^2+^, 0.16 mM dNTP, 0.25 μM of each primer, and deionized water up to 20 μL. The PCR protocol involved initial denaturation (94 °C, 5 min) and then 40 cycles consisting of steps: denaturation (94 °C, 30 s), annealing (56 and 58 °C, 40 s), elongation (72 °C, 55 s), and final elongation (72 °C, 5 min). The obtained DNA probes were purified with the QIAquick PCR Purification Kit (Qiagen, Hilden, Germany), sequenced, and labeled by random priming with the HexaLabel DNA Labeling Kit (Fermentas, Waltham, MA, USA) and radioisotope 50 μCi [α-32P]-dCTP. Finally, probes were hybridized with the narrow-leafed lupin nuclear genome BAC library, as previously described by Książkiewicz et al. (2013). Verification of positive hybridization signals was performed by PCR and Sanger sequencing with gene-specific primers (Table 5).

### 3.3. Estimating GS and PEPC Sequence Variant Numbers

To estimate the number of *GS* and *PEPC* sequence variants in the *L. angustifolius* genome, droplet digital PCR (ddPCR) was performed with the use of the Bio-Rad QX200 Droplet Digital PCR System (Bio-Rad, Hercules, CA, USA). The set of *GS* and *PEPC* specific primers was anchored in the most conserved gene regions among legume plants with well-established sequence data. A gene described as a single copy in the narrow-leafed lupin genome, namely aspartate aminotransferase *(AAT*) [31,91], was used as the reference in the ddPCR experiment. A series of *L. angustifolius* genomic DNA dilutions, ranging from 0.125 to 2.0 ng/μL, were used as templates in ddPCR reactions containing 2× QX200 ddPCR EvaGreen Supermix (Bio-Rad, Hercules, CA, USA), 200 nM gene-specific primers, and 50–80 nM AAT-specific primers. The final volumes of ddPCR reactions (20 μL), together with 70 μL of droplet generation oil, were placed in DG8 Cartridges, partitioned into droplets by the QX200 Droplet Generator (Bio-Rad, Hercules, CA, USA) and transferred into 96-well plates. The ddPCR protocol involved initial denaturation (95 °C for 5 min), followed by 40 cycles consisting of steps: denaturation (95 °C, 30 s), annealing (60 and 61 °C, 30s), elongation (72 °C, 45 s), and final elongation (72 °C, 45 s). The fluorescence was read on the QX200 Droplet Reader (Bio-Rad, Hercules, CA, USA). On average, 17,000 droplets were analyzed per 20 μL PCR. The data analysis was performed with QuantaSoft droplet reader software (Bio-Rad, Hercules, CA, USA) that incorporates the Poisson distribution algorithm. Supplementary to this analysis, recently released *L. angustifolius* sequencing data (Lupin Express: annotated gene set cds v1.0 and genome sequence GCA_001865875.1) were screened in order to identify all variants of analyzed genes.

### 3.4. Characterizing GS1, GS2, and PEPC Gene Variants, as well as Their Corresponding L. angustifolius Genome Regions

Whole BAC insert sequencing was performed by the Miseq platform (Illumina, San Diego, CA, USA) in a paired-end 2 × 250 bp approach (Genomed, Warsaw, Poland).

The narrow-leafed lupin genome scaffold assembly v1.0 (GCA_000338175.1) and genome pseudochromosome assembly v1.0 (GCA_001865875.1) were used to obtain *GS* and *PEPC* gene variant sequences, not represented in BAC clones, and to establish their positions in the genome. The BLAST algorithm was optimized for highly similar sequences: e-value cut-off, 1 × 10^−20^; word size, 28; match/mismatch scores, 1/-2; and gap costs, linear.

The obtained BAC clone insert sequences and narrow-leafed lupin scaffold fragments corresponding to the narrow-leafed lupin genome regions carrying *GS1, GS2*, and *PEPC* genes (average length of 100 kb) were subjected to computational characterization of repetitive content and gene coding sequences. Repetitive elements were annotated and masked using RepeatMasker Web Server version 4.0.3 (search engine, cross_match; speed/sensitivity, slow; DNA source, *Arabidopsis thaliana*) and supplemented with the CENSOR tool accessed via the Genetic Information Research Institute (sequence source, Viridiplantae; force translated search; mask pseudogenes).

Gene prediction was performed using FGENESH [92] with *G. max* as a reference species. Functional annotation of predicted coding sequences was performed with the use of the BLAST algorithm (e-value cut-off, 1 × 10^−10^ word size, 28; match/mismatch scores, 1/-2; and gap costs, linear). The obtained *GS1, GS2*, and *PEPC* gene structures were visualized and compared in Geneious software v 10.1 (http://www.geneious.com). The results of functional annotation were subsequently used for gene density (genes/kbp) calculation.

### 3.5. Positioning GS1, GS2, and PEPC in NLL Pseudochromosomes

To assign particular *GS* and *PEPC* gene variants to narrow-leafed lupin pseudochromosomes, in silico mapping was performed. *L. angustifolius* genome sequence data (GCA_001865875.1) and the latest version of the species genetic map were used [11,21]. The BLAST algorithm was optimized as follows: e-value cut-off, 1 × 10^−20^; word size, 28; match/mismatch scores, 1/-2; and gap costs, linear. Moreover, previously developed molecular markers anchored within *GS1* (036L23_3, 047P22_3, 087N22_2, and 059J08_3) and *PEPC* (064J15_5, 067C07_2, and 131K15_5_3) gene sequences were incorporated into this study [31].

### 3.6. Describing Local Genome Rearrangements Harboring GS and PEPC Loci

To identify and describe local genome rearrangements and microsynteny patterns in regions carrying *GS* and *PEPC* genes in narrow-leafed lupin and nine Fabaceae species, *L. angustifolius* BAC sequences with a repetitive content were masked by RepeatMasker and Censor [93] and subjected to comparative mapping. The following genome sequences were used: *A. duranensis* (Peanut Genome Project accession V14167, http://www.peanutbase.org*), A. ipaensis* (Peanut Genome Project accession K30076, http://www.peanutbase.org) [6], *C. cajan* [7] (project PRJNA72815, v1.0), *C. arietinum* [8] (v1.0 unmasked, http://comparative-legumes.org), *G. max* [9] (JGI v1.1 unmasked, http://www.phytozome.net), *L. japonicus* [10] (v2.5 unmasked, http://www.kazusa.or.jp), *M. truncatula* [12] (strain A17, JCVI v4.0 unmasked, http://www.jcvi.org/medicago), *P. vulgaris* (v0.9, DOE-JGI, and USDA-NIFA; http://www.phytozome.net) [13], and *V. radiata* [14] (GenBank/EMBL/DDBJ accession JJMO00000000). The CoGe BLAST algorithm [94] was used to perform sequence similarity analyses with the following parameters: e-value cutoff, 1 × 10^−20^; word size, 8; gap existence cost, 5; gap elongation cost, 2; and nucleotide match/mismatch scores, 1/−2. Microsyntenic blocks were visualized using the Web-Based Genome Synteny Viewer [95] and Circos [96].

### 3.7. Phylogenetic Reconstruction of the Plant Species Tree

The reference genome sequences were gathered from Phytozome [97], NCBI/RefSeq [98], and Ensembl/Plants [99] databases. A full list of genomes and respective sources is available in Appendix A.

For species tree reconstruction, a set of conserved homologs were selected with conditional reciprocal BLAST (CRB-BLAST) [100] against the Ensembl/Plants version of the *A. thaliana* representative proteome (longest encoded protein at each coding locus) with default settings. Singular loci with over 95% representation as single-copy orthologs over all the analyzed species were selected for species tree reconstruction, yielding a total of 29 loci. The alignment of representative protein sequences for each orthologous locus was obtained with MAFFT-LINSi v 7.310 [101], and a 70% occupancy threshold was used to filter the alignments with trimal, while simultaneously back translating to underlying codons with the *-backtrans* option provided in trimal [102]. All alignments were concatenated and partitioned analysis was conducted on the basis of this joint supermatrix. The list of all loci (by *A. thaliana* reference locus) and the respective evolutionary models used can be found in Appendix A.

An approximate species tree was reconstructed with IQTREE v 1.5.5 [103]. Optimal model selections [104] were carried out using IQTREE’s built-in capabilities (MFP option). Ultrafast bootstrap approximation [105] was used to assess the topology based on a 3000 iteration threshold (convergence was reached in 104 iterations).

### 3.8. Determining GS1, GS2, and PEPC Gene Families Evolutionary Patterns

Sequences were gathered with independent BLASTP (2.6.0) searches of each included plant genome (including non-legume reference genomes; full list included as Appendix A) and the July 2017 version of the UniProt/SwissProt (The UniProt Consortium 2017) golden standard database. The resulting hits were filtered based on the maximum 1 × 10^−20^ expectation value threshold and the minimum 40% coverage of at least one of the lupin homologs sequenced during the experimental phase of the project (sequences obtained from sequenced BAC clones: 047P22, 087N22, 036L23, 059J08, 064J15, 067C07, and 131K15 used as queries). Supervised clustering was then conducted in a procedure analogous to that described in our earlier work [46] and the sequences were compared against each other with USEARCH (UBLAST v8.1.1831 search with e-value threshold 1 × 10^−10^) [106]. Finally, the pairwise relationships (e-values post log-transformation) were used to cluster the sequences with MCL [107] at multiple inflation threshold values. The optimal value of the inflation threshold was selected as 1.4, based on the averaged values of the silhouette width [108], which is a cluster quality measure independent of predefined class labels. The largest clusters, which contained all of the GS/PEPC hits found in SwissProt, were processed further. SwissProt sequences were initially kept for purposes of alignment/filtering, but were discarded for final phylogenetic tree reconstruction/reconciliation.

In order to filter out assembly errors, heavily truncated partial genes, and/or pseudogenes, additional criteria were used. All accepted sequences were aligned with MAFFT v7.310 and preprocessed with OD-seq [109]. OD-seq uses a gap-based distance metric to filter out outliers with significantly different gap patterns compared to the rest of alignment. Prior to assessment, a round of trimming was carried out with trimal, based on a very permissive 1% gap threshold (parameter choice resulting in retaining sequences longer than average). All discarded sequences can be found in Appendix A. The *PEPC* sequence from *Archaeoglobus fulgidus* and *GS* sequence from *Rhizobium meliloti* were initially used to guide rooting (pruned prior to reconciliation), and both coding sequences were selected on the basis of respective SwissProt records.

During GS analysis, a singular, a previously established [110] sequence for *L. japonicus* was introduced in lieu of seemingly duplicated loci on the sixth pseudochromosome of the draft genome (Lj6g3v0410480/Lj6g3v0410490; both corresponding to C-terminal part of the full coding sequence). A comparison of the *L. japonicus* pseudochromosome and reference sequence of the previously cloned region, has shown that likely misassembly or recombination has affected the region, so the reference UniProt sequence was used in downstream analyses.

During PEPC analyses, sequences from the *Volvox carteri* NCBI/RefSeq genome were used in lieu of Phytozome version due to the higher gene model quality. Additionally, available sequences from *Chlamydomonas reinhardtii* were obtained through UniProt/SwissProt records (and corresponding GenBank entries), as the current reference genome does not contain full-length gene models corresponding to either PEPC1 or PEPC2.

Phylogenetic inference was conducted analogous to the species tree reconstruction described above (IQ-TREE, optimal model selection, ultrafast bootstrap approximation). Codon-based models and coding sequences were used in order to obtain a better resolution of recent bipartitions. The SCHN05 model [111] with a free-rate model of site heterogeneity [112] was selected in both cases (GS:SCHN05+R6, PEPC:SCHN05+R8). Based on the rule of parsimony, reconstructions with the least amount of inferred duplications/losses (minimum cost of optimal reconciliation based on DTL-RANGER [113] reconciliations of species/gene trees, with disabled horizontal transfer events) were chosen. Notably, this resulted in the selection of codon-based nucleotide alignments over protein sequences and the abandonment of alignment trimming for gene tree reconstruction. The visualization of optimal reconciliation was carried out with custom scripts in the Python/ETE2 environment based on the built-in ETE2 reconciliation procedure and DTL-RANGER results [114].

### 3.9. Selection Pressure Analysis

Pairwise selection pressure parameters, including Ka (the number of nonsynonymous substitutions per nonsynonymous site), Ks (the number of synonymous substitutions per synonymous site), and Ka/Ks ratios, were calculated in DnaSP 5 [115]. To follow the topologies of the trees, the branch-site test of positive selection was performed in PAML4 [116]. Two models were considered: a null model, in which the foreground branch might have different proportions of sites under neutral selection to the background (i.e., relaxed purifying selection), and an alternative model, in which the foreground branch might have a proportion of sites under positive selection. The hypothesis of positive selection was verified by the likelihood ratio test (alternative vs. null model) and *p*-value under a Chi-square distribution and one degree of freedom (maximum *p*-value threshold of 0.05 was used). Sites under positive selection for foreground lineages were predicted by naive empirical Bayes and Bayes empirical Bayes [117] (a minimum posterior probability threshold of 0.95 was used). Both analyses were based on the same alignments as those used for phylogenetic inference; however, codons present in less than 30% of sequences from a particular clade were removed (Appendix A).

## 4. Conclusions

*GS* and *PEPC* genes were shown to have had a complex history, with bacterial-type PEPCs emerging as those best suited for future phylogenetic inquiries into relationships between divergent legumes.Legume *GS* and *PEPC* genes evolved by both ancestral legume-wide and more recent lineage-specific WGDs. Descendants of these duplications have been retained in the majority of lineages and have sustained typical gene structures, implying differences in carbon/nitrogen metabolism due to regulatory rather than mechanistic changes.Legume *PEPC* and *GS* gene sequences were highly conserved by significant purifying selection. Tentative traces of positive selection can only be inferred in several branches and point to single residues, outside of the core set involved in ligand binding.Monocot family members of the *GS* gene family might be more ancient than dicot ones, stemming from the selective culling of duplicates predating the separation of both lineages.The general patterns of lineage-specific duplications suggest that sub-functionalization and/or regulatory rewiring played a large role in shaping the extant carbon and nitrogen primary metabolic pathways in some lineages (*L. angustifolius*, *L. japonicus*, and *G. max*).

## Figures and Tables

**Figure 1 ijms-21-02580-f001:**
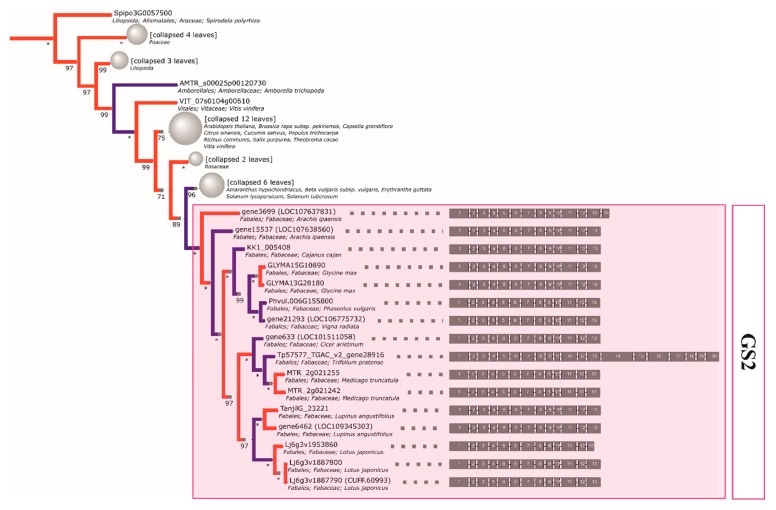
The reconstructed phylogeny of plant plastid GS isoenzyme (*GS2*) genes. A collapsed phylogeny tree was used in order to highlight Fabaceae family relationships.

**Figure 2 ijms-21-02580-f002:**
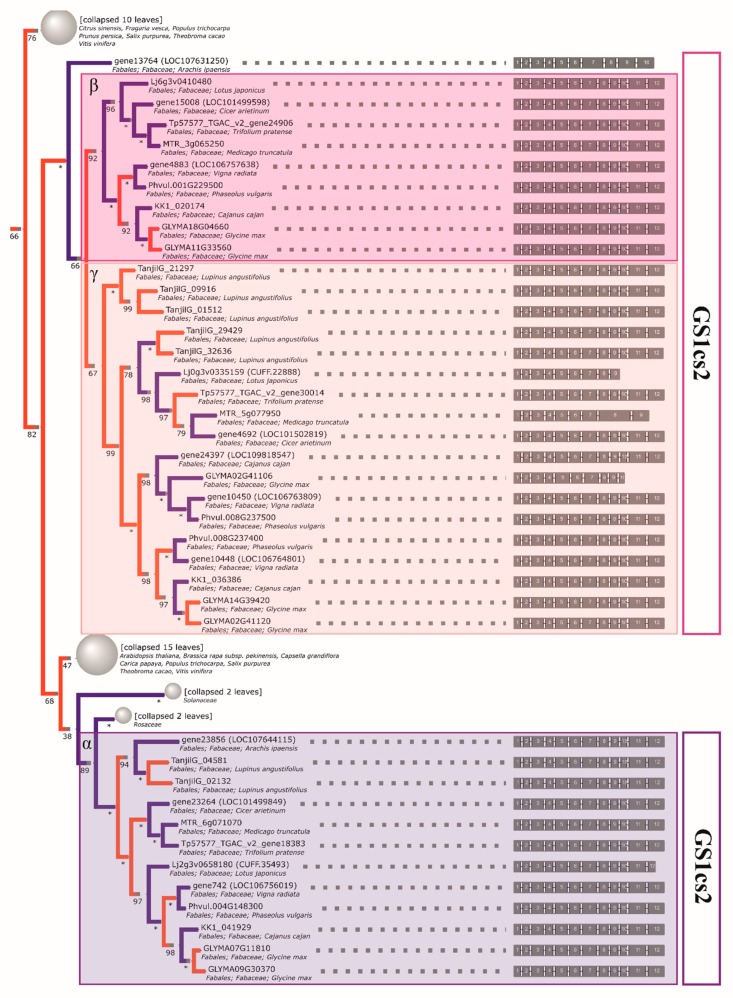
The reconstructed phylogeny of plant cytosolic GS isoenzyme (*GS1*) genes. A collapsed phylogeny tree was used in order to highlight Fabaceae family relationships.

**Figure 3 ijms-21-02580-f003:**
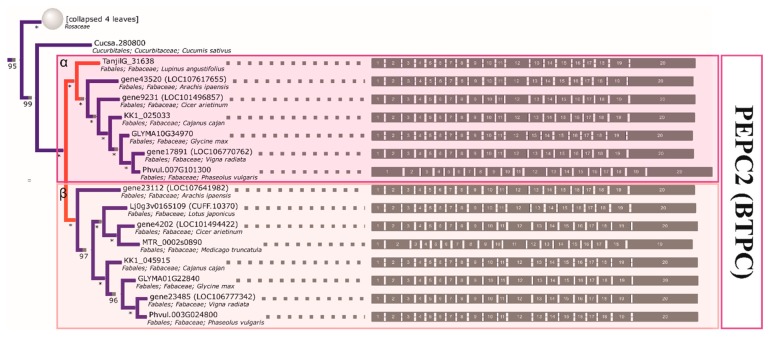
The reconstructed phylogeny of plant *PEPC* genes. A collapsed phylogeny tree was used in order to highlight Fabaceae family relationships.

**Figure 4 ijms-21-02580-f004:**
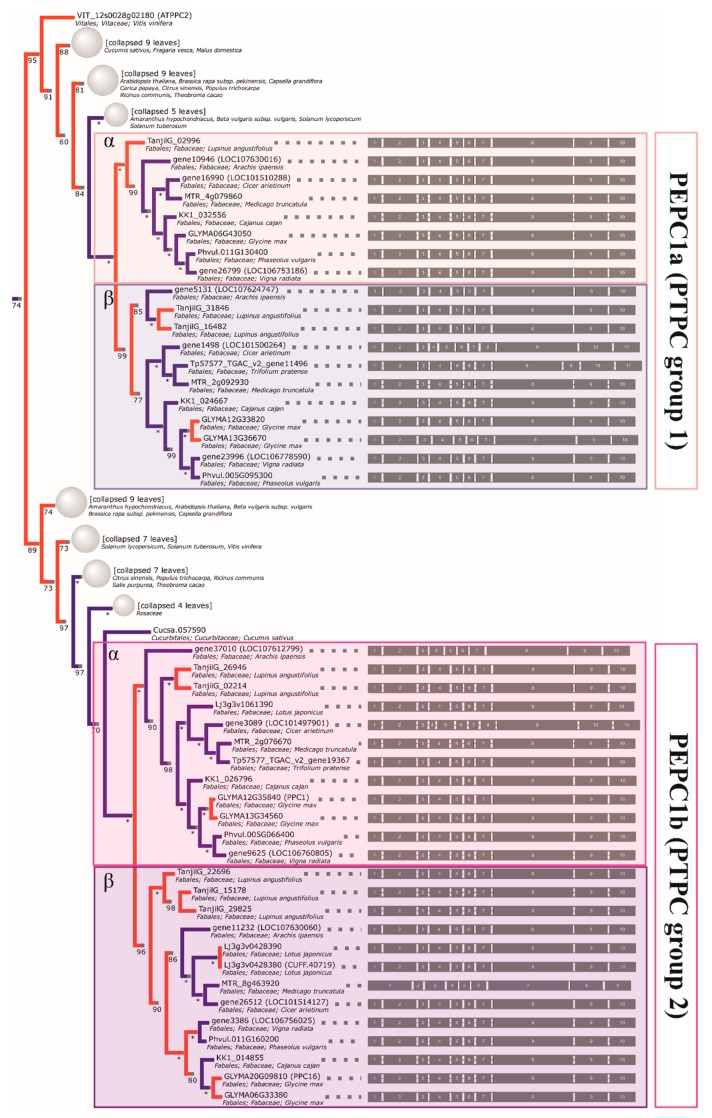
The reconstructed phylogeny of plant *PEPC* genes. A collapsed phylogeny tree was used in order to highlight Fabaceae family relationships.

**Figure 5 ijms-21-02580-f005:**
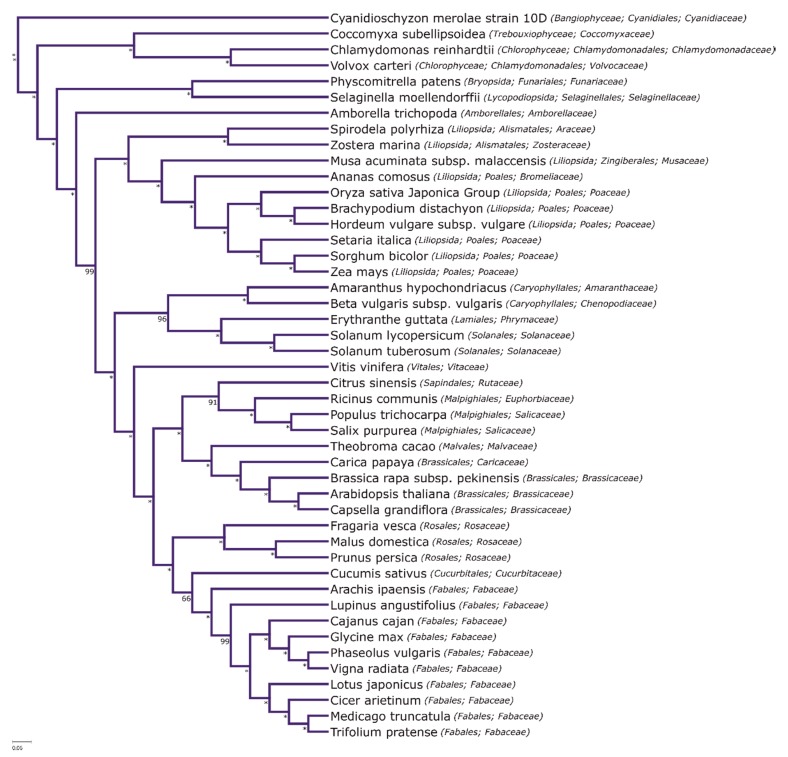
Maximum likelihood codon-based phylogenetic species tree of 46 reference plant genomes, based on 29 putative single-copy orthologs.

**Figure 6 ijms-21-02580-f006:**
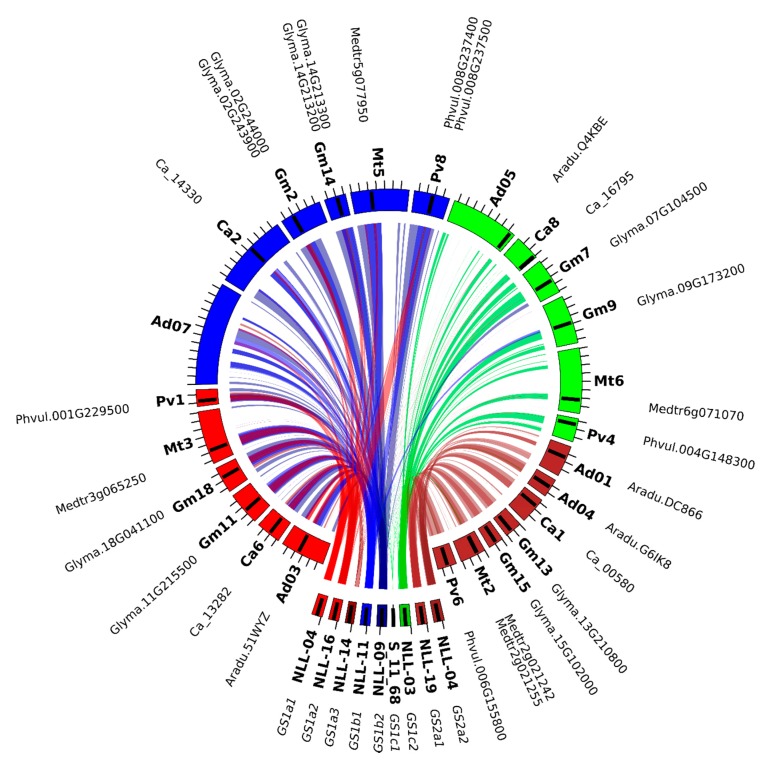
Collinearity links matching narrow-leafed lupin linkage groups and the legume reference genome carrying *GS* genes. NLL*—*narrow-leafed lupin linkage group, Pv*—P. vulgaris*, Mt*—M. truncatula*, Gm*—G. max*, Ca*—C. arietinum*, and Ad*—A. duranensis*.

**Figure 7 ijms-21-02580-f007:**
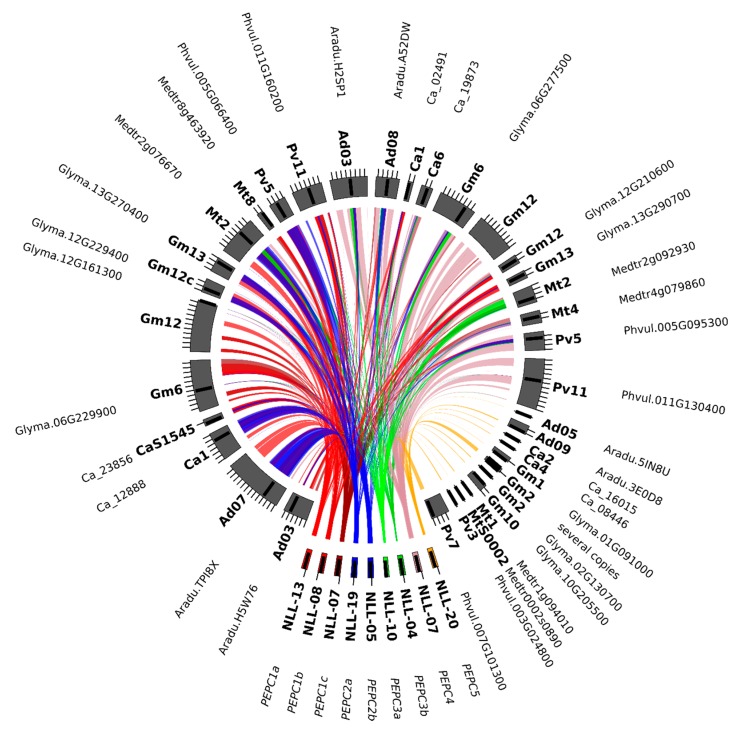
Comparative mapping and phylogenetic inference of legume *PEPC* genes. Syntenic patterns revealed for narrow-leafed lupin genome regions and corresponding regions of legume chromosomes. NLL*—*narrow-leafed lupin linkage group, Pv*—P. vulgaris*, Mt*—M. truncatula*, Gm*—G. max*, Ca*—C. arietinum*, and Ad*—A. duranensis*.

**Table 1 ijms-21-02580-t001:** Characterization of *Lupinus angustifiolius* bacterial artificial chromosomes (BACs)/scaffolds carrying glutamine synthetase (*GS*) and phosphoenolpyruvate carboxylase (*PEPC*) sequences, including anchoring genes to the scaffolds and narrow-leafed lupin linkage groups (NLLs), cytogenetic marker representation, and repetitive content analysis within selected scaffolds. NLL—narrow-leafed lupin linkage group, RE—repetitive element, and CDS—coding sequence.

Gene Variant	Gene ID		BAC nb	Scaffold nb	NLL nb	Cyto marker	GC%	% RE	RE (bp)	RE type	CDS nb
	Lupin Express ID	GenBank ID									
**GS1a1**	**Lup21297**	**gene6261**	**047P22**	**4_25**	**4**	**047P22_5**	**33.1**	**8.58**	8584	Ty1/Copia	12
**GS1a2**	Lup001512	gene27466	087N22	106	16	087N22_2	32.94	15.63	15635	TY1/Copia; Gypsy/DIRS1; DNA transposons	10
**GS1a3**	Lup009916	gene24502	-	192	14	-	36.11	10.54	9282	Ty1/Copia; Gypsy/DIRS1; DNA transpozons	17
**GS1b1**	Lup029429	gene 19431	036L23	73	11	036L23_3	33	0	0	-	15
**GS1b2**	Lup032636	gene17555	059J08	94_15	9	059J08_3	32.43	0.17	174	Ty1/Copia	16
**GS1c1**	Lup002132	gene34907	-	11_68	UN	-	30.56	9.19	2621	Ty1/Copia; DNA transposons	3
**GS1c2**	Lup04581	gene4422	-	13	3	-	31.89	7.2	7202	Ty1/Copia; DNA transposons	15
**GS2a1**	Lup023221	gene31805	-	45_213	19	-	33.88	9.96	9963	Ty1/Copia; Gypsy/DIRS1; DNA transpozons	12
**GS2a2**	no	gene6462	-	186	4	-	32.66	7.73	7732	Ty1/Copia; Gypsy/DIRS1; DNA transpozons	12
**PEPC1a**	Lup022696	gene23490	064J15	437	13	-	34.25	8.85	8852	Ty1/Copia; Gypsy/DIRS1	13
**PEPC1b**	Lup029825	gene15450	-	74_10	8	-	32.28	1.88	1879	Ty1/Copia	13
**PEPC1c**	Lup015178	gene12998	-	274	7	-	32.36	1.17	1169	Ty1/Copia	14
**PEPC2a**	Lup002214	gene31196	067C07	110_41	19	067C07_2	32.41	3.63	3634	Ty1/Copia; Gypsy/DIRS1	14
**PEPC2b**	Lup26946	gene9184	131K15	59_19	5	131K15_5_3	33.21	6.49	6748	Ty1/Copia; Gypsy/DIRS1	14
**PEPC3a**	Lup031846	gene18605	-	9_1	10	-	33.97	5.17	1628	Ty1/Copia	3
**PEPC3b**	Lup016482	gene7147	-	296	4	-	32.76	15.64	15641	Ty1/Copia; DNA transposon	5
**PEPC4**	Lup002996	no	-	12_32	7	-	33.76	8.64	8644	Ty1/Copia; Gypsy/DIRS1	16
**PEPC5**	Lup031638		-	88_60	20	-	32.73	8.93	8933	Ty1/Copia; Gypsy/DIRS1; DNA transposons	10

**Table 2 ijms-21-02580-t002:** Summary of major glutamine synthetase clades traced to the ancestral legume genome (monophyletic, support over 90%).

GS Subset	Legume Clade	Taxon	Locus Tag (NCBI: Gene Locus ID ^1^)
**GS2**	dalbergioids	*Arachis ipaensis*	gene15537 (LOC107638560), gene3699 (LOC107637831)
	genistoids	*Lupinus angustifolius*	TanjilG_23221, gene6462 (LOC109345303)
	IRLC	*Cicer arietinum*	gene633 (LOC101511058)
		*Medicago truncatula*	MTR_2g021242, MTR_2g021255
		*Trifolium pratense*	Tp57577_TGAC_v2_gene28916
	milletioids	*Cajanus cajan*	KK1_005408
		*Glycine max*	GLYMA13G28180, GLYMA15G10890
		*Phaseolus vulgaris*	Phvul.006G155800
		*Vigna radiata*	gene21293 (LOC106775732)
	robinioids	*Lotus japonicus*	Lj6g3v1887790 (CUFF.60993), Lj6g3v1887800, Lj6g3v1953860
**GS1cs1**	dalbergioids	*Arachis ipaensis*	gene13764 (LOC107631250)
	genistoids	*Lupinus angustifolius*	TanjilG_32636, TanjilG_29429, TanjilG_09916, TanjilG_01512, TanjilG_21297
	IRLC	*Cicer arietinum*	gene15008 (LOC101499598), gene4692 (LOC101502819)
		*Medicago truncatula*	MTR_3g065250, MTR_5g077950
		*Trifolium pratense*	Tp57577_TGAC_v2_gene24906, Tp57577_TGAC_v2_gene30014
	milletioids	*Cajanus cajan*	gene24397 (LOC109818547), KK1_036386, KK1_020174
		*Glycine max*	GLYMA02G41106, GLYMA02G41120, GLYMA11G33560, GLYMA14G39420, GLYMA18G04660
		*Phaseolus vulgaris*	Phvul.001G229500, Phvul.008G237400, Phvul.008G237500
		*Vigna radiata*	gene10448 (LOC106764801), gene10450 (LOC106763809), gene4883 (LOC106757638)
		*Lotus japonicus*	Lj0g3v0335159 (CUFF.22888), Lj6g3v0410480
**GS1cs2**	dalbergioids	*Arachis ipaensis*	gene23856 (LOC107644115)
	genistoids	*Lupinus angustifolius*	TanjilG_02132, TanjilG_04581
	IRLC	*Cicer arietinum*	gene23264 (LOC101499849)
		*Medicago truncatula*	MTR_6g071070
		*Trifolium pratense*	Tp57577_TGAC_v2_gene18383
	milletioids	*Cajanus cajan*	KK1_041929
		*Glycine max*	GLYMA07G11810, GLYMA09G30370
		*Phaseolus vulgaris*	Phvul.004G148300
		*Vigna radiata*	gene742 (LOC106756019)
	robinioids	*Lotus japonicus*	Lj2g3v0658180 (CUFF.35493)

^1^ Where a locus tag is not available (gene designated as the NCBI reannotation only), the NCBI Gene database ID is given in the parentheses, prefixed with LOC.

**Table 3 ijms-21-02580-t003:** Summary of major phosphoenolpyruvate carboxylase clades traced to the ancestral legume genome (monophyletic, support over 90%).

PEPC Subset	Legume Clade	Taxon	Locus Tag (NCBI:Gene Locus ID ^1^)
**PEPC1a**	dalbergioids	*Arachis ipaensis*	gene10946 (LOC107630016), gene5131 (LOC107624747)
	genistoids	*Lupinus angustifolius*	TanjilG_02996, TanjilG_31846, TanjilG_16482
	IRLC	*Cicer arietinum*	gene1498 (LOC101500264), gene16990 (LOC101510288)
		*Medicago truncatula*	MTR_2g092930, MTR_4g079860
		*Trifolium pratense*	Tp57577_TGAC_v2_gene11496
	milletioids	*Cajanus cajan*	KK1_024667, KK1_032556
		*Glycine max*	GLYMA06G43050, GLYMA12G33820, GLYMA13G36670
		*Phaseolus vulgaris*	Phvul.005G095300, Phvul.011G130400
		*Vigna radiata*	gene23996 (LOC106778590), gene26799 (LOC106753186)
**PEPC1b**	dalbergioids	*Arachis ipaensis*	gene11232 (LOC107630060), gene37010 (LOC107612799)
	genistoids	*Lupinus angustifolius*	TanjilG_15178, TanjilG_29825, TanjilG_22696, TanjilG_02214, TanjilG_26946
	IRLC	*Cicer arietinum*	gene26512 (LOC101514127), gene3089 (LOC101497901)
		*Medicago truncatula*	MTR_2g076670, MTR_8g463920
		*Trifolium pratense*	Tp57577_TGAC_v2_gene19367
	milletioids	*Cajanus cajan*	KK1_014855, KK1_026796
		*Glycine max*	GLYMA06G33380, GLYMA12G35840 (PPC1), GLYMA13G34560, GLYMA20G09810 (PPC16)
		*Phaseolus vulgaris*	Phvul.005G066400, Phvul.011G160200
		*Vigna radiata*	gene3386 (LOC106756025), gene9625 (LOC106760805)
	robinioids	*Lotus japonicus*	Lj3g3v0428380 (CUFF.40719), Lj3g3v0428390, Lj3g3v1061390
**PEPC2**	dalbergioids	*Arachis ipaensis*	gene23112 (LOC107641982), gene43520 (LOC107617655)
	genistoids	*Lupinus angustifolius*	TanjilG_31638
	IRLC	*Cicer arietinum*	gene4202 (LOC101494422), gene9231 (LOC101496857)
		*Medicago truncatula*	MTR_0002s0890
	milletioids	*Cajanus cajan*	KK1_025033, KK1_045915
		*Glycine max*	GLYMA01G22840, GLYMA10G34970
		*Phaseolus vulgaris*	Phvul.003G024800, Phvul.007G101300
		*Vigna radiata*	gene17891 (LOC106770762), gene23485 (LOC106777342)
	robinioids	*Lotus japonicus*	Lj0g3v0165109 (CUFF.10370)

^1^ Where a locus tag is not available (gene designated as the NCBI reannotation only), the NCBI Gene database ID is given in the parentheses, prefixed with LOC.

**Table 4 ijms-21-02580-t004:** Normalized leaf expression level of *GS* and *PEPC* genes in a *L. angustifolius* recombinant inbred line (RIL) mapping population (83A:476 x P27255) [85].

Gene	Accession	Mean Expression in RIL Population	Min Expression Value in RIL Population	Max Expression Value in RIL Population	Expression SD
*GS1a1*	Lup021297	43.1	20.4	74.1	16.4
*GS1a2*	Lup001512	13.8	4.9	32.2	5.2
*GS1a3*	Lup009916	11.5	3.6	43.7	5.0
*GS1b1*	Lup029429	0.3	0.0	1.4	0.3
*GS1b2*	Lup032636	2.6	0.4	6.0	1.2
*GS1c1*	Lup002132	0.1	0.0	0.9	0.2
*GS1c2*	Lup004581	187.5	117.6	426.6	52.4
*GS2a1*	Lup023221	516.2	365.3	739.7	80.3
*GS2a2*	-	-	-	-	-
*PEPC1a*	Lup022696	17.0	8.1	31.1	4.0
*PEPC1b*	Lup029825	0.5	0.0	2.4	0.5
*PEPC1c*	Lup015178	65.0	44.5	87.4	8.7
*PEPC2a*	Lup002214	0.0	0.0	0.5	0.1
*PEPC2b*	Lup026946	0.1	0.0	0.9	0.2
*PEPC3a*	Lup031846	10.5	4.7	16.1	2.5
*PEPC3b*	Lup016482	51.0	33.0	94.2	11.2
*PEPC4*	Lup002996	1.7	0.0	4.1	0.9
*PEPC5*	Lup031638	11.9	1.7	28.7	4.8
*HEL*	Lup023733	3.0	0.4	7.4	1.2
*TUB*	Lup021845	78.4	35.3	113.1	15.2

SD*—*standard deviation; HEL and TUB*—*reference genes.

**Table 5 ijms-21-02580-t005:** Gene-specific primers used for the probe amplification and verification of positive hybridization signals.

Probe Name	PCR Primer Sequence	Length (bp)	T*
GS	GS_F: GTTGGTCCCTCTGTTGGAATCTCTG GS_R: ATAAGCAGCAATGTGCTCATTGTGTCTC	571	56
PEPC	PEPC_F: AAAGATGTTAGGAATCTTCACATGCTGCAAGA PEPC_R: GGGGCATATTCACTTGTTGGGTTCAGT	643	58

T**—*melting temperature.

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
