# Peer review of "A Tale of Two Families: Whole Genome and Segmental Duplications Underlie Glutamine Synthetase and Phosphoenolpyruvate Carboxylase Diversity in Narrow-Leafed Lupin (Lupinus angustifolius L.)"

_ijms, 2020, doi:10.3390/ijms21072580_

Round 1

Reviewer 1 Report

   Review of IJMS-735506

Tale of two families – whole genome and segmental duplications underlie glutamine synthetases and phosphoenolpyruvate carboxylases diversity in narrow-leafed lupin

Katarzyna Czyż, Michał Książkiewicz, Grzegorz Koczyk, Anna Szczepaniak, Jan Podkowiński, Barbara Naganowska

The authors used concensus sequences of known legume glutamine synthase (GS) and

phosphoenolpyruvate carboxylase (PEPC) genes to design probes to identify Bacterial Artificial Chromosomes containing the narrow-leafed lupin (Lupinus angustifolius) orthologs of these genes. Using this approach they identified and sequenced BACs containing copies of both genes, but using digital droplet PCR they estimated that they had missed copies of each gene so they filled in the gaps by probing the annotated L. angustifolius genome. Using these combined approaches they identified 9 GS and 9 PEPC genes. They then characterized the structure and evolution of these genes. Finally, they used these sequences together with the sequences from many other legumes and other outliers to construct phylogenetic trees describing the evolution of the GS and PEPC gene families in legumes, and, by inference, the evolution of the legumes themselves.

Overall, the impression I get is that this is two separate papers cobbled together. The first is about the isolation and sequencing of the lupin GS and PEPC genes, and the second is about the evolution of the legume GS and PEPC gene families, and, by inference, the evolution of the legumes themselves. Although these studies clearly interconnect, the link is rough and the quality of the two parts is very different.

My largest criticism is why was this study even necessary? What new information does it provide, and how can this information be used? It seems that the new data used in these analyses are the sequences of the L. angustifolius GS and PEPC genes, yet they were only slight additions to the pre-existing data used for the analyses in sections 3.4 through 3.8. What additional information did they provide? How can this information be used?

Another major criticism is why it was even necessary to identify and sequence BACs containing lupin GS and PEPC genes, since the complete genomic sequence was already available and since they were able to use it to identify genes that they missed by their approach? Did they find genes that were missed in the annotated genome? Were there other reasons to doubt that the annotated L. angustifolius genome contained all GS and PEPC genes? Since this portion of the study consumes much of the materials and methods and initial results it should be justified, especially since sections 3.4 through 3.8 of the results and discussion primarily use published data for the reported analyses.

A final criticism is the absence of a “conclusions” section. What new information was learned from this study? What new information does it provide about plant evolution in general, or evolution of the legumes? How can it be used for lupin improvement?

More specific concerns:

The English must be extensively revised, since there are mistakes in nearly every sentence. I recommend revision by a native English speaker.

Please give the latin binomial for narrow-leafed lupin in title and abstract.

Captions to figures and tables should provide enough information to understand them without reading the text. Therefore, all abbreviations and acronyms must be explained in the caption

Please be sure to clarify new sequence data generated in this study and what was available previously. (e.g line 202).

Lines 259-260 must be rewritten to clarify that used a GS probe to identify BACs harboring GS genes and a PEPC probe to identify BACs harboring PEPC genes.

Lines 259-275 are confusing and must be rewritten for clarity. My perception is that they identified seven GS1 genes, two GS2 genes and nine PEPC genes.

Line 287: all of the studied species are legumes!

Line 303: caption to table 2 needs to explain the table better. What are NLL? When they say %RE, do they mean that the gene model itself, eg. GS1a2, is 15.63% RE, or that the BAC clone is 15.63% RE?

Line 319: Table 1 should be Table 2.

Lines 328-332 need to be rewritten for clarity. Are they saying that these RE were contained within the gene model, or in the DNA flanking the gene model?

Author Response

Please see the attachememt.

Reviewer 2 Report

In this study Czyż et al presented a complex characterisation of two narrow-leafed lupin gene families, glutamine synthetase (GS) and phosphoenolpyruvate carboxylase (PEPC). they combine a comparative analysis of gene structures and synteny-based approach together with phylogenetic reconstruction and reconciliation of gene family and species history in order to examine events underlying the extant diversity of both families. they showed the impact of duplications on the initial complement of analysed gene families within genistoid clade and posit that the function of duplicates has been largely retained.

The work as well as the results and discussions are well written and in some places the discussions are very substantial. the work shows a large previous study. Usually the methodologies are described superficially but this is not the case. On the contrary, in this work also the materials and methods section is described in a very detailed way. For me in this version the work can be accepted.

Author Response

We would like to thank for such a great note.

Round 2

Reviewer 1 Report

The authors have satisfactorily answered my concerns. There are still many minor problems with the English, but these do not affect the ability to understand the manuscript.

I therefore recommend publication after fixing the English.
